# Unveiling the Dark Side of Negative Behaviors Among Nurses and Their Implications in Workforce Well-Being and Patient Care: A Scoping Review

**DOI:** 10.3390/healthcare13162079

**Published:** 2025-08-21

**Authors:** Nuno Santos, Rita Barahona, Paulo Cruchinho, Elisabete Nunes

**Affiliations:** 1Nursing Research, Innovation and Development Centre of Lisbon (CIDNUR), Nursing School of Lisbon, Avenida Prof. Egas Moniz, 1600-190 Lisbon, Portugal; ralmeida1@campus.esel.pt (R.B.); pjcruchinho@esel.pt (P.C.); enunes@esel.pt (E.N.); 2Hospital da Luz de Lisboa, Avenida Lusíada 100, 1500-650 Lisbon, Portugal; 3Unidade Local de Saúde de São José, Rua José António Serrano, 1150-199 Lisbon, Portugal; 4Nursing Administration Department, Nursing School of Lisbon, Avenida Prof. Egas Moniz, 1600-190 Lisbon, Portugal

**Keywords:** nurses, workplace violence, negative behaviors, scoping review, nursing administration research

## Abstract

**Introduction**: Negative behaviors in nursing undermine well-being, erode team cohesion, and jeopardize patient safety. Rooted in systemic stressors—workload, emotional strain, and power imbalances—they have far-reaching effects on job satisfaction and care quality. **Objective**: To systematically map the scientific evidence on negative behaviors among nurses in healthcare organizations. **Methods**: A scoping review was conducted using five databases: CINAHL, MEDLINE, Scopus, Psychology & Behavioral Sciences Collection, and RCAAP (for grey literature). The review followed the Joanna Briggs Institute methodology and PRISMA-ScR reporting guidelines. Two independent reviewers conducted data extraction and synthesis. **Results**: Eighteen studies published between 2017 and 2024 met inclusion criteria from an initial pool of 88 references. Eleven thematic domains emerged: (1) the cycle of violence; (2) victims profile; (3) perpetrator profile; (4) negative behaviors spectrum; (5) negative behaviors prevalence; (6) risk predictors; (7) protective predictors; (8) impact of negative behaviors on nurses; (9) impact of negative behaviors on patients; (10) impact of negative behaviors on healthcare organizations; (11) organizational strategies and the role of the nurse managers. **Conclusions**: The findings highlight the multidimensional nature of negative behaviors and the variability in how they are defined and assessed. This review highlights the need for conceptual clarity and standardized tools to address negative behaviors in nursing. Nurse managers, as key organizational agents, play a critical role in fostering psychological safety, promoting ethical leadership, and ensuring accountability. System-level strategies that align leadership with organizational values are essential to protect workforce well-being and safeguard patient care.

## 1. Introduction

Negative behaviors in the nursing profession have been widely documented as an ongoing and global issue [1]. Such behaviors significantly disrupt both formal and informal interpersonal processes that are vital to organizational functioning and healthcare delivery [2]. Rather than isolated instances of individual misconduct, these behaviors are increasingly understood as structural and systemic, deeply embedded in organizational cultures and sustained over time. The literature consistently characterizes them as “endemic” with “deep historical roots”, and as a “global phenomenon” that transcends cultural, institutional and geographic boundaries [1]. 

The emergence of systematic inquiry into these behaviors dates to the 1980s, with the work of Leymann, who developed the Leymann Inventory of Psychological Terror (LIPT) to categorize 45 forms of hostile workplace behavior. His studies identified healthcare workers—particularly nurses—as especially vulnerable due to rigid professional hierarchies, emotionally demanding environments, and asymmetric power dynamics [3].

Despite decades of empirical research, a coherent and unified conceptual framework remains lacking. The literature offers a variety of overlapping terminologies to describe these behaviors, such as unprofessional attitudes and improper or unacceptable social behaviors [4,5,6,7]. Different frameworks conceptualize negative workplace behaviors in diverse ways. The International Labour Organization (ILO) adopts a broad definition encompassing abuse of power, verbal and sexual harassment, incivility, and ostracism [8]. In contrast, Nemeth et al. focus specifically on nursing, framing these behaviors as harmful relational manifestations with consequences at both the individual and organizational levels [9]. While the ILO perspective emphasizes structural and legal dimensions, Nemeth’s framework highlights interpersonal dynamics within healthcare teams [8]. This review adopts Nemeth’s operational definition due to its conceptual specificity and alignment with the nursing context [9].

The persistence of these behaviors is further explained by theoretical frameworks grounded in organizational sociology. Power-conflict theory, as advanced by Coser and Dahrendorf, posits that hierarchical structures and unequal power distributions inherently generate latent tensions. When unaddressed, these tensions can escalate into aggression and dysfunctional dynamics [10,11].

Building on this foundation, Murray’s concept of silent epidemics adds a critical layer of interpretation by addressing the perpetuation of negative behaviors through processes of normalization, institutional neglect, and underreporting [12]. Together, these frameworks reveal how such behaviors arise from systemic power imbalances and become culturally ingrained as tolerated, even expected, elements of professional socialization [1].

Current scientific evidence describe the existence of four types of negative behavior in the healthcare sector: type (1), criminal intent towards the organization; type (2), from patients, family members, or carers towards the organization’s multidisciplinary team; type (3), inside the organization’s multidisciplinary team; and type (4), involving an aggressor who has a personal relationship with the victim but not with the organization [13]. This study will only focus on negative behaviors that occur among nurses (type 3).

These intra-organizational behaviors can manifest horizontally, between peers of equal hierarchical status (e.g., staff nurse to staff nurse), or vertically, involving unequal power relations (e.g., nurse manager to staff nurse or vice versa) [14].

It is well established that the vitality of any organization fundamentally depends on its human capital [5]. Therefore, healthcare organizations are inherently complex systems ripe for conflict [6,7]. Conflicts often arise from perceived incompatibilities between goals, emotions, or perspectives, and are amplified by the division of tasks, shared responsibilities, and operational interdependence. In healthcare settings, these dynamics are further exacerbated by emotionally charged environments, limited resources, and complex relational networks. Such conditions are intrinsic to organizational life and have traditionally been viewed as threats to team cohesion and productivity [4,5,6,7]. Adverse nursing practice environments are consistently associated with burnout, decreased job satisfaction, and higher turnover intentions among nurses [15,16,17,18,19,20,21,22].

Nurse managers, embedded within the organizational structure, are instrumental in mediating the relationship between negative behaviors and the quality of the nursing work environment. Effective nurse management can foster a culture of psychological safety, mutual respect, and team cohesion [15,16,17]. Conversely, the absence of formal reporting mechanisms, combined with fear of retaliation, enables the perpetuation of harmful behavior and undermines efforts at organizational accountability [23].

Although prior studies have examined specific aspects of negative behaviors in nursing, the literature remains fragmented, with no comprehensive synthesis that consolidates typologies, contributing factors, and measurement approaches into a unified conceptual framework. This gap limits the development of standardized assessment tools and evidence-based organizational interventions.

To address this, we conducted a scoping review focused on negative behaviors among nurses within healthcare organizations, identifying key domains, measurement approaches, and knowledge gaps relevant to research, policy, and clinical practice. 

This effort was guided by the following research question: What is the scientific evidence on negative behaviors among nurses in healthcare organizations? The primary objective of this study is to systematically map the available scientific evidence on negative behaviors among nurses in healthcare organizations.

## 2. Methods

### 2.1. Design

We conducted a scoping review, a form of knowledge synthesis that employs a systematic and iterative approach to identify and synthetize an existing and emerging body of literature on negative behaviors among nurses. Although a scoping review protocol by Almost et al. was identified, which aimed to explore behaviors within workplace relationships, no scoping reviews were found that focused exclusively on negative behaviors among nurses [24]. 

The rationale for conducting a scoping review lies in its ability to comprehensively map existing evidence and examine the breadth and diversity of available knowledge. In contrast to systematic reviews, which are typically designed to inform evidence-based clinical guidelines, this approach accommodates a wider range of study designs and information sources—beyond peer-reviewed literature—offering a structured yet flexible framework to assess the current state of evidence and identify gaps for future inquiry [25,26,27].

### 2.2. Protocol and Registration

Before starting the literature research, this scoping review was brought about by the scoping review protocol registered with the Open Science Framework under the following domain: https://osf.io/y4bng/?view_only=0fd4fd321f4940bd919e7ecea1bc8190 (accessed on 3 May 2025).

### 2.3. Eligibility Criteria

This scoping review was guided by Joanna Briggs Institute (JBI) PCC framework, which is designed to support the development of inclusion criteria and structure the research question for scoping reviews. The PCC acronym stands for population, concept, and context [25,26,27]. For this specific review: population refers to the group of interest; concept focuses on the key phenomenon under investigation; and context defines the setting or environment relevant to the review. The combination of these mnemonics brought force to the review question (“What is the scientific evidence on negative behaviors among nurses in healthcare organizations?”).

The following criteria were included for this review:

Population: studies that focused exclusively on licensed registered nurses employed in public or private healthcare organizations, regardless of their specific nursing category, years of clinical experience, age, gender, academic background, or employment relationship. Only studies in which the sample consisted entirely of nurses were included; studies involving mixed samples (e.g., nurses and other healthcare providers or ancillary staff) were excluded. Nurse managers were included when their roles were explicitly clinical or when they were part of broader nursing teams in practice settings. In contrast, studies focusing solely on nurse faculty, academic nurses, educators, nursing students, nursing assistants, and practical/vocational nurses were excluded, as these groups operate in educational or support roles that differ significantly from the clinical and organizational dynamics experienced by registered nurses in direct patient care.

Concept: studies that address one or more negative behaviors or that discuss the concept of negative behavior. The concept of negative behavior providing the basis for this review consists of a set of work attitudes towards other professionals having the potential to negatively impact individuals and organizations [9]. Within the set of negative behaviors presented above, this scoping review will focus on negative behaviors such as bullying, incivility, ostracism, lateral/horizontal violence, or colleague violence [9,14].

Context: all studies reporting or discussing results from healthcare organizations, covering primary care, as well as hospital and long-term care, regardless of the type of institutions either public or private.

Types of Resources: Qualitative, quantitative, and mixed-approach studies and systematic reviews that fitted the defined criteria were considered for inclusion. Although grey literature was considered eligible, no studies were retrieved from RCAAP during the search process.

Given the diversity of terminology historically associated with these phenomena, the time frame for the review was established between 2017 and 2024 to capture the period following a conceptual shift in the literature. This shift was marked by seminal contributions such as Nemeth et al. The influence of this work can be observed in a series of subsequent studies that adopted and further developed this terminology and framework [9].

This scoping review included studies published in English, Portuguese, and Spanish, with no restrictions on their geographical location. The remaining studies published in other languages were not accountable for this review. Those languages’ selection was based on the review team’s language proficiency to ensure accurate interpretation and data extraction. 

This framework provides a structured approach for determining eligibility criteria and ensures that the review maintains a broad yet focused exploration of the existing literature (Table 1).

### 2.4. Information Sources and Research Strategies

An initial exploratory search was carried out in the CINAHL and MEDLINE databases (accessed via EBSCOhost) to identify pertinent keywords and indexing terms. These databases were selected due to their recognized relevance and comprehensive coverage in nursing-related literature reviews [28].

This scoping review was conducted according to the Preferred Reporting Items for Systematic Review and Meta-Analyses extension for Scoping Reviews (PRISMA-ScR) guidelines [25,26,27,29] (see Appendix A). 

A comprehensive search strategy was implemented across five databases: CINAHL, MEDLINE, Psychology & Behavioral Sciences Collection, SCOPUS, and RCAAP. The selection of these databases was based not only on their thematic relevance to the research topic but also on their recognized authority and breadth in indexing high-quality, peer-reviewed literature in the fields of health sciences and nursing—thus supporting a robust and comprehensive evidence base [28]. 

The previously identified keywords were utilized in rigorous combinations using Boolean operators and truncations, with search equations tailored to the indexing structure of each database. The research was conducted on 16 December 2024. The complete search terms used in the five different databases are found in Table 2.

Grey literature was not included due to no relevant results being retrieved in RCAAP. This is reflected in the PRISMA flowchart diagram (Figure 1). 

### 2.5. Selection Process

All identified references were organized and imported into the Mendeley reference management software (version 1.19.8), where duplicate records were removed. The remaining studies were then imported into the Rayyan Enterprise and Rayyan Teams+ application (version 1.6.0), for screening [30].

The screening process followed a methodological framework comprising three distinct phases by the same two independent reviewers (NS and RB). In the initial step (1), the literature was screened based on titles and abstracts against the eligibility criteria defined in the review protocol. Studies deemed potentially relevant proceeded to the second phase (2), which involved full-text screening. In the final phase (3), the selected studies were consolidated, and their reference lists were examined using a manual snowballing strategy to identify additional relevant literature [25,26,27,31]. When abstracts lacked sufficient detail to assess eligibility, the full texts were retrieved and thoroughly reviewed to determine inclusion. 

Any discrepancies were consensually settled between the reviewers and, when necessary, with the mediation of a third reviewer [25,26,27,31]. The identification, screening, and inclusion of studies were documented using the PRISMA-ScR flowchart diagram, ensuring methodological rigor and transparency consistent with the exploratory objectives of a scoping review [30].

Consistent with the JBI framework for scoping reviews, no critical appraisal of methodological quality was conducted. While this approach aligns with the primary goal of mapping the breadth of scientific evidence, it limits the ability to assess the internal validity of the included studies. This tradeoff was accepted to ensure inclusiveness and capture a wide spectrum of conceptual and contextual data [25].

### 2.6. Data Collection Process

To extract the data, an Excel file was developed based on the model proposed by the JBI [31].

Data extraction was conducted independently by two reviewers, in line with the methodological approach recommended by JBI guidelines. It was then systematized in a standardized table developed for the purpose of this study [31]. This table was built according to the review’s objective and the respective review question [27], including key information from each source: author(s); year of publication; country of origin; study purpose; sample size (when applicable); type of negative behavior analyzed; data collection instruments (when applicable); methodology adopted; main results; and limitations relevant in the framework of this scoping review.

The discrepancies between the two reviewers were solved during the selection process and data collection process; therefore, consultation with a third reviewer was not necessary.

## 3. Results

### 3.1. Screening Results

A total of 88 records were identified through database searches, with 20 being removed due to duplication. Out of 68 records, 32 were excluded after analyzing the titles, resulting in 36 articles for screening by abstract. In the second phase, 9 articles were eliminated based on their abstracts, leading to the selection of 27 articles for full-text evaluation. Of these, 14 articles were excluded because they did not meet the eligibility criteria, due to the inadequacy of the population studied (n = 7), the concept addressed (n = 6), or because they were instrument validation studies (n = 1). A review of the list of references (hand searching) of the included studies and expert recommendations resulted in five additional articles. Thus, a total of 18 records met the eligibility criteria and were included in the synthesis. These steps are synthesized in Figure 1, following the PRISMA flowchart diagram.

The 18 studies included were published between 2017 and 2024 (44.44% studies published during 2017–2020 and 55.55% published during 2021–2024), and all were written in English (n = 18, 100%). Figure 2 refers to the characteristics of the included studies. In short, these studies were conducted in 8 countries: 14 of them (77.78%) were conducted in high-income countries—United States [32,33,34,35,36,37], Italy [38], Canada [39,40], China [41,42,43] and Australia [1,44]; while the remaining 4 studies (22.22%) were conducted in middle-income and low-income countries—Turkey [45], Ghana [46], Egypt [47], and Malaysia [48].

Out of the included studies, 11 (61.11%) adopted quantitative approaches [32,33,35,38,39,41,42,43,45,46,47], 3 (16.67%) were qualitative studies [1,40], and 4 (22.22%) were integrative reviews [34,36,37,44]. The samples ranged from 13 participants [1] to large groups of more than 1900 nurses [47], totaling 8500 participants for this study. This methodological and geographical heterogeneity offers, therefore, a comprehensive view of negative behaviors (Figure 2).

Negative behaviors were analyzed using varied terminologies across the included studies. The term bullying was the most frequently reported, appearing in seven studies (38.89%), followed by incivility in three studies (16.67%), and ostracism in one study (5.56%). Three studies (16.7%) employed broader labels, with two (11.11%) referring to negative workplace behaviors and one (5.56%) to deviant workplace behaviors. Additionally, five studies (27.78%) used peer-specific terminology: lateral violence in two studies (11.11%), horizontal violence in two studies (11.11%), and colleague violence in one study (5.56%). To provide a visual summary of the prevalence of the most reported negative behaviors among nurses across the included studies, a pie chart was constructed (Figure 3).

To assess negative behaviors, a total of 14 data collection instruments were identified. The Negative Acts Questionnaire-Revised (NAQ-R) was the most frequently reported and utilized, with 6 out of the 18 studies (33.33%) employing this tool in their research [32,35,38,39,45,46].

Regarding the settings, the included studies encompassed various healthcare contexts: eight studies (44.44%) were conducted in a variety from various healthcare settings [32,33,35,36,39,43]; and seven studies (38.89%) were conducted in hospital environments [1,33,40,41,42,44,45], among which four studies (22.22%) focused specifically on acute care settings [1,40,41,44] and one on a public hospital (5.56%) [48].

Table 3(a,b) displays the charted data from each selected study, aligned with the review objective and research question.

### 3.2. Analysis of the Selected Studies

As mentioned, the aim of this scoping review is to systematically map the available scientific evidence on negative behaviors among nurses in healthcare organizations. To meet this objective, 18 studies were included that address the expression of these behaviors within nursing teams, generating the following domains: the cycle of violence; victims profile; perpetrator profile; negative behaviors spectrum; negative behaviors prevalence; risk predictors; protective predictors; impact of negative behaviors on nurses; impact of negative behaviors on patients; impact of negative behaviors on healthcare organizations; and organizational strategies and the role of nurse managers.

#### 3.2.1. The Cycle of Violence

Negative behaviors in nursing often involve fluid roles of victim, perpetrator, and bystander, shaped by contextual pressures and power dynamics. As Edmonson & Zelonka [34] note, these roles shift frequently, especially in high-stress environments.

Bambi et al. [38] found that 32.4% of respondents had experienced both victimization and perpetration, illustrating how individuals may replicate harmful behaviors in response to prior exposure. Hawkins et al. [1] and Krut et al. [40] describe these dynamics as cyclical, where retaliation—such as withholding information or reducing collaboration—emerges as a learned response, reinforcing dysfunctional team patterns.

Trépanier et al. [39] emphasize the role of bystanders, whose silence or inaction can normalize aggression and perpetuate unequal power structures. Their passive complicity often sustains a permissive environment for mistreatment.

These findings underscore the opportunity for meaningful cultural transformation when organizations recognize the systemic roots of negative behaviors and invest in leadership development, team empowerment, and bystander engagement as drivers of a healthier and more supportive work environment [39], as illustrated in Figure 4.

#### 3.2.2. Victims Profile

Findings from the included studies reveal recurring patterns regarding the profiles of nurses most frequently exposed to negative behaviors. Early-career nurses emerged as a particularly vulnerable group, identified in 33.33% of the studies (6 out of 18), often described in the literature through the expression “nurses are eating their young” [1,35,36,37,40,44]. These professionals commonly report emotional fragility, low self-esteem, and feelings of isolation, especially in workplace cultures marked by an “us versus them” mentality [1,37,44].

Newly hired staff, referenced in 5.56% of the studies (1 out of 18), also appear at increased risk, particularly during the integration phase in unfamiliar teams [1]. Similarly, nurses occupying lower hierarchical positions were reported in 11.11% of the studies (2 out of 18) as more frequently exposed to negative behaviors [1,46]. Only one study (5.56%) highlighted nurses with higher academic qualifications or experience as common targets, potentially due to perceived threats to informal power dynamics within teams [38].

Gender also emerged as a relevant variable. Despite women representing over 89% of the samples across the studies included, 16.67% of studies (3 out of 18) reported that female nurses experienced greater exposure to negative behaviors [31,41,48], suggesting an intersection between gender and organizational vulnerability. 

#### 3.2.3. Perpetrator Profile

The characteristics of individuals who perpetrate negative behaviors in nursing settings are multifaceted. Power—whether formal, informal, perceived, or contested—consistently emerges as a central element influencing interpersonal dynamics and the occurrence of such behaviors. In 1 out of the 18 included studies (5.56%), perpetrators were reported to occupy positions of authority within the nursing hierarchy, suggesting that power asymmetries may contribute to the perpetuation of negative behaviors [46].

Individual traits have also been explored. Mansor et al. identify trait anger as a potential contributing factor, although no association was found between negative affectivity and the enactment of negative behaviors, suggesting that emotional predispositions alone may not explain such conduct [48].

Certain behaviors are framed by perpetrators as professionally justified and described by Hawkins et al. as “tough love” or a means to uphold clinical standards and patient safety [1]. Age-related dynamics are observed in both directions, with reports of negative behaviors from older nurses toward younger colleagues and vice versa [31].

Psychological factors have also been reported. Edmonson & Zelonka suggest that some perpetrators may experience low self-confidence and perceive competent peers as threats [34].

Avoidance-based coping strategies, such as psychological withdrawal or rigid separation between personal and professional spheres, may limit team cohesion and indirectly sustain negative behaviors [40].

Table 4 provides a synthesis of the key characteristics associated with victims and perpetrators.

#### 3.2.4. Negative Behaviors Spectrum

The included studies demonstrate considerable conceptual diversity regarding negative behaviors in nursing settings. Mansor et al. define negative behaviors as unethical actions that violate organizational norms and harm individuals or the work environment. These behaviors range from low-severity forms (e.g., absenteeism, tardiness) to high-severity forms (e.g., theft, harassment, insubordination) and vary by periodicity (sporadic vs. systematic) and by intensity (passive vs. active) [48].

Incivility (addressed in 16.67% of the studies) is described as a low-intensity behavior with ambiguous intent to harm, disrupting interpersonal dynamics. It includes discourtesy, a dismissive tone, and the omission of basic expressions of politeness [33,42]. When persistent and intentional, it may overlap with bullying [36].

Ostracism (addressed in 5.56% of the studies) refers to a subtle social exclusion, such as avoiding interaction, restricting access to common spaces, or relocating individuals to peripheral areas, leading to perceived marginalization [47].

Colleague violence (addressed in 5.56% of the studies) encompasses psychological hostility between colleagues of the same hierarchical level, affecting team cohesion and collaboration [45]. Horizontal violence and lateral violence (addressed in 22.22% of the studies) refer to peer-directed aggression, either overt (e.g., verbal abuse) or covert (e.g., exclusion) [38,40,43].

Bullying (addressed in 38.89% of the studies) is consistently defined by three core elements: intentionality, repetition, and power imbalance. Manifestations include verbal abuse, social exclusion, and spreading rumors. Typologies include person-related, work-related, and physical bullying. Some definitions introduce temporal thresholds, such as weekly episodes over a minimum of six months [34,35,37,38,41,46].

The diversity of terminology found across studies highlights a valuable opportunity to strengthen conceptual coherence in the field. Advancing toward standardized definitions and validated assessment tools will enhance comparability and the quality of future research [45]. As a step in this direction, Table 5 summarizes the types of negative behaviors identified in the included studies, outlining their definitions and key characteristics.

#### 3.2.5. Negative Behaviors Prevalence

The prevalence of negative behaviors among nurses varies widely, shaped by conceptual definitions, data collection instruments, and contextual factors. Horizontal and colleague violence are notably recurrent. Krut et al. [40] reported that 59.1% of nurses experienced horizontal violence within a six-month period, while Bambi et al. [38] found colleague violence affecting 47% of respondents.

Bullying specifically has been extensively documented as a persistent occupational hazard. Prevalence rates range from 27% to 40%, as observed by Sauer & McCoy [35] and Anusiewicz et al. [37]. Bambi et al. [38] noted that 35.8% of nurses faced negative behaviors, with 42.3% of these exposed to weekly bullying episodes for at least six months. Similar patterns were reported by Lu et al. (30.6%) [41] and Trépanier (40%) [39]. Olender [29] further highlighted the intensity of these behaviors, with 26.3% of nurses reporting daily bullying and 35.9% experiencing it weekly.

#### 3.2.6. Risk Predictors

Organizational antecedents such as high workloads, resource constraints, staff fatigue, and ineffective communication have been associated with increased exposure to incivility and bullying [36]. Excessive workload was also found to be a relevant determinant, though its effect may be attenuated by social support and recognition [39]. Inadequate leadership and suboptimal management, especially in planning and resource distribution, also contribute to the emergence of conflict and interpersonal tension [37]. 

Through logistic analysis, Bambi et al. identified several predictors of negative interactions: working day shifts, exposure to peer-directed hostility, and previous engagement in negative behaviors [38]. 

Geographical settings may also play a role. In rural or remote areas, lower turnover rates and sustained interpersonal ties may increase susceptibility to exclusion particularly among newly integrated staff. These dynamics seemed to have been reinforced during the COVID-19 pandemic [1].

#### 3.2.7. Protective Predictors

Working in hospital settings was associated with a lower prevalence of bullying, suggesting that institutional structure may offer a protective effect [38]. Greater professional autonomy was inversely related to incivility [34], and strong work ethic predicted lower levels of negative behaviors [47].

Constructive coping strategies—such as open communication and proactive task adjustment—were associated with reduced interpersonal tension [45]. Social support from colleagues, mentors and external networks emerged as a key factor in mitigating psychological impact [1].

Self-care practices, including physical activity, boundary setting, and distancing from conflict, were reported to enhance emotional regulation [40]. However, their effectiveness appears dependent on broader organizational support. Resilience was noted to be beneficial but insufficient in the absence of engaged leadership and structured psychosocial resources [35].

Institutional policies promoting psychological safety and clear response frameworks were identified as essential to strengthening the overall capacity for conflict prevention and management [32,38].

To illustrate these risks and protective determinants, Table 6 was developed.

#### 3.2.8. Impact of Negative Behaviors on Nurses

Negative behaviors are consistently associated with physical and psychological harm among nurses. Early effects include emotional distress, insecurity, cognitive impairment, and low self-confidence [33,35,40,45,47]. Persisting exposure contributes to symptoms such as chronic fatigue, sleep disturbance, gastrointestinal complaints, and headaches [35,40,45].

Among the most critical outcomes is burnout (addressed in 50% of the included studies), a progressive and debilitating syndrome resulting from sustained occupational stress. It is typically characterized by emotional exhaustion, depersonalization, and a reduced sense of personal accomplishment. Burnout impairs clinical reasoning, emotional regulation, and relational engagement, ultimately affecting both individual functioning and team dynamics [33,35,37]. 

Moreover, Bambi et al. reported that 59% of nurses subjected to negative behaviors experienced psychophysical harm [38]. In advanced stages, prolonged exposure may evolve into severe psychopathological states such as major depression, post-traumatic stress disorder, and psychache—an intense form of psychological pain driven by humiliation, despair, and emotional breakdown [39,41,46]. Lu et al. found that negative behaviors were significantly associated with suicidal ideation and suicide attempts [41]

These adverse effects often extend into nurses’ personal lives, increasing the risk of interpersonal conflict, substance misuse, and persistent psychological rumination [31,35,45]. As reported by Hawkins et al., the emotional toll of workplace mistreatment often extended beyond the clinical setting, with participants describing severe psychological spillover into their personal lives. This was vividly illustrated through qualitative narratives in which some nurses referred to their experiences as “living hell” [1].

#### 3.2.9. Impact of Negative Behaviors on Patients

The psychological burden experienced by nurses impairs clinical vigilance, judgment, and communication, particularly in high-acuity settings [35,37]. Anusiewicz et al. observed associations between exposure to mistreatment and increased rates of medication errors and patient falls. Emotional exhaustion and cognitive overload compromise decision making and delay interventions [37].

Hostile work environments weaken communication among team members, diminishing collective responses during emergencies [35]. As emphasized by Sauer & McCoy, these dynamics reduce collaborative capacity and elevate the risk of adverse events [35]. Ultimately, poor practice environments compromise patient safety, satisfaction, and dignity [33].

#### 3.2.10. Impact of Negative Behaviors on Healthcare Organizations

At the organizational level, negative behaviors undermine retention, efficiency, and performance. Toxic work environments contribute to elevated attrition rates and intra-organizational transfers [1,34,36,37,38,39,44,46]. Bambi et al. reported that 21.9% of nurses experiencing mistreatment intended to exit the profession and 20% sought reassignment [38].

Although some nurses compensate for negative behaviors with increased productivity, these efforts are often unsustainable and lead to emotional exhaustion [45]. Xiaolong et al. found that incivility negatively impacts polychronicity, defined as the individual tendency to engage in multiple tasks simultaneously. This exposure weakens the positive association between polychronicity and job satisfaction, leading even highly adaptable nurses to experience emotional fatigue and disengagement [42].

Challenging work environments may also entail considerable financial implications. Edmonson & Zelonka estimated that issues related to workplace dynamics, including turnover and decreased care quality, can result in annual costs ranging from USD 4 to 7 million per hospital in the United States. In addition, team disruption and staffing fluctuations can affect continuity of care, influence performance metrics, and impact the institution’s public perception [34].

Table 7 summarizes the reported impacts of negative behaviors across multiple levels, including nurses, patients, and healthcare organizations.

#### 3.2.11. Organizational Strategies and the Role of Nurse Managers

The reviewed studies indicated that organizational context influences both the prevalence of negative behaviors and the capacity of nurse managers to address them effectively. Leadership practices do not operate in isolation but are shaped by institutional culture, available resources, and systemic support mechanisms.

Organizations that promote autonomy, emotional support, and open communication are consistently associated with lower levels of incivility and interpersonal conflict [33]. Smith et al. found that such environments reported fewer episodes of mistreatment [33], while Olender observed that staff exposed to engaged and supportive nurse managers reported reduced hostility [32].

In contrast, authoritarian or top-down leadership models appear to contribute to workplace tension. Farrell reported that hierarchical structures may reinforce power imbalances and limit open dialogue, reducing psychological safety and hindering conflict resolution [36].

Several studies highlighted the importance of leadership-led interventions supported at the organizational level. Hawkins et al. emphasized the role of nurse managers in advancing respectful and safe work environments, noting the need for multi-level strategies anchored in leadership [44]. Farrell proposed a framework that includes early identification of incivility, elimination of behavioral triggers, professional development, and the use of cognitive rehearsal techniques [36].

Policy-oriented approaches were described by Edmonson & Zelonka, who outlined steps for implementation including: recognizing the problem, mitigating aggravating factors such as overload and fatigue, training leaders in communication, enforcing behavioral standards, and promoting accountability among staff [34].

Bambi et al. recommended continuing education, anonymous reporting systems, access to occupational psychologists, and rotation of team compositions as organizational strategies to reduce interpersonal strain [38].

Despite these efforts, some barriers to implementation were noted across studies. These include fear of retaliation, limited confidence, managerial inaction, time constraints, and the perceived indispensability of certain staff members [29,31,34,36,39,43]. 

Table 8 was developed to present strategies to prevent and address negative behaviors among nurses.

## 4. Discussion

The present scoping review synthesized findings from 18 studies, highlighting the multidimensional nature of negative behaviors among nurses.

The eleven domains identified in this review—ranging from typologies and prevalence to protective predictors and organizational strategies—should not be interpreted in isolation. Rather, they constitute an interdependent framework in which individual, relational, and systemic variables interact recursively to shape the manifestation and persistence of negative behaviors in nursing environments.

These behaviors span a continuum of incivility, ostracism, bullying, and lateral violence and are deeply embedded in organizational and interpersonal dynamics. The results confirm the persistence and complexity of these phenomena, aligning with existing literature that positions them as systemic rather than isolated issues [49,50,51].

One of the most salient findings is the cyclical nature of mistreatment within nursing teams, where individuals alternate between roles of victim, perpetrator, and bystander [38]. These roles are often shaped by exposure to organizational stressors, socialization processes, and power asymmetries. Several studies described how such patterns become normalized over time, particularly in environments that tolerate aggression under the guise of disciplinary action or “tough love” [1,52]. This aligns with the concept of “organizational Darwinism”, wherein emotional detachment and competitiveness are inadvertently reinforced by hierarchical structures [53,54].

Professional seniority does not provide uniform protection. While novice and early-career nurses remain the most frequent targets [1,38,49,51,55,56], several studies reported that mid-career professionals and nurse managers may also experience negative behaviors, often related to informal power struggles or perceived threats to group cohesion [1,37,41,44]. The circularity of roles, where those previously victimized may later perpetuate similar behaviors, illustrates how negative behaviors can perpetuate themselves across generations of professionals [41,54]. 

The role of bystanders emerged as a critical factor in either reinforcing or mitigating harmful behaviors. Passive or avoidant bystanders contribute to normalization of mistreatment, while active or suppressive bystanders may help to reshape team norms [39,50]. These findings underscore the importance of fostering collective accountability and cultivating workplace cultures where interventions are encouraged and supported.

Another important observation was the inconsistency in terminology and conceptual frameworks across the studies. Studies used multiple overlapping terms, despite key differences in severity, intentionality, and duration [32,38,40,41,44,46,57]. This lack of standardization complicates measurement and intervention [49,55,58]. These challenges echo those previously reported, which identified up to 13 distinct terms describing similar workplace phenomena [59]. 

Contrary to isolated findings suggesting hospital settings may offer protection [38], the broader evidence shows that negative behaviors are more likely to occur in hospital-based contexts [57,60,61,62], especially in high-pressure units. Emergency departments [60,63], operating rooms [62], surgical wards [64,65], oncology units [66,67], and correctional facilities [68] were consistently associated with increased exposure to mistreatment. 

Our review identified several risk factors associated with negative behaviors, which are also reflected in the existing literature. For example, maladaptive coping strategies such as avoidance and substance use have been highlighted by João & Portelada [23], while Jang et al. [51] emphasized the role of low emotional intelligence and strong meritocratic beliefs in reinforcing such behaviors within performance-oriented cultures.

Burnout was identified as both a key outcome and a mediating mechanism. Defined by Maslach et al. [69], it was consistently linked to bullying, chronic stress, and disengagement [33,34,35,36,37,38,39,44,46,70,71]. João et al. found nearly 70% of bullied nurses are at moderate or high burnout risk [72]. Ultimately, burn-on was described as a state of persistent emotional exhaustion masked by continued functionality [73]. 

Institutional commitment to zero-tolerance policies appears to be increasingly framed in the literature as essential for long-term workforce sustainability [34,35,38,44,51,52]. Supportive leadership—underpinned by emotional intelligence, psychological safety, and conflict management skills—emerged as a key protective factor [51]. Interventions like coaching, cognitive rehearsal, and nonviolent communication showed promise [35,74], yet several authors noted that many managers lack these competencies, often defaulting to avoidance or punitive strategies that exacerbate toxic dynamics [1,34].

Hopefully, our study unveiled multiple protective factors closely aligned with international evidence emphasizing the buffering effect of supportive environments [33,47]. The iceberg model of competencies offers a valuable lens: while surface-level skills like clinical expertise are often prioritized, hidden competencies—emotional intelligence, ethical reasoning, conflict mediation—are decisive in shaping culture [75]. 

To break these cycles, organizations must move beyond surface-level interventions and cultivate environments where psychological safety and collaborative competence are valued as much as clinical expertise [75]. These findings suggest that addressing negative workplace behaviors requires multi-level strategies. 

Certain expressions cited in this review (e.g., “a living hell”, “nurses eating their young” [1]) were directly quoted from original studies to faithfully illustrate participants’ experiences and the language used in qualitative data. While vivid, these phrases were interpreted within a neutral analytical framework and do not reflect editorial bias.

Ultimately, power-conflict theory frames negative behaviors as outcomes of unresolved tensions and systemic asymmetries [10,11]. Addressing them requires aligning leadership with organizational structures. While nurse managers are pivotal in fostering safe environments, their effectiveness depends on the systems that support them. Sustainable change stems not from isolated actions but from integrated strategies that promote psychological safety, accountability, and relational care. In doing so, we move from “us versus them” to a culture of “we care together.”

## 5. Limitations and Future Prospects

This review presents several limitations that may affect the possibility of generalizing its findings. The selection of four databases and the inclusion of studies published exclusively in English, Portuguese, or Spanish may have introduced language and publication bias, potentially omitting relevant evidence from other linguistic or regional contexts. 

The included studies used diverse and, at times, overlapping terminology to describe negative behaviors (e.g., bullying, incivility, lateral violence). While this conceptual heterogeneity complicates direct comparison, it also reflects the current fragmentation in the field and reinforces the need for standardized definitions. 

Although validated tools such as the Nursing Incivility Scale and the Negative Behaviors in Healthcare Survey are commonly used in the field, studies employing these instruments may have been underrepresented in our results due to limitations in indexing and search string specificity. In this review, we deliberately adopted the term “nurses” alongside complementary descriptors of negative behaviors to ensure thematic focus. Nonetheless, we acknowledge that incorporating broader or truncated terms (e.g., “nursing”, “nurs*”, “registered nurs*”) could have enhanced the sensitivity of the search. Future reviews should consider such refinements to maximize retrieval scope and inclusiveness.

As noted in the methodology, and in line with JBI guidance, no critical appraisal was performed, which although appropriate for a scoping review limits assessment of evidence rigor. The findings should therefore not be assumed to derive solely from high-quality studies; acknowledging this enhances transparency and scientific integrity.

Future research should prioritize longitudinal, quasi-experimental, and mixed-methods designs to evaluate the effectiveness of leadership-based and educational interventions. Emphasis should be placed on measurable outcomes such as burnout, turnover, missed nursing care, and patient safety.

Special attention should be given to vulnerable populations—namely newly graduated nurses—and to the implementation of safe reporting mechanisms. The integration of qualitative approaches, such as interviews and ethnographic methods, will enable a more comprehensive understanding of the organizational and cultural dimensions that sustain negative behaviors.

## 6. Conclusions

This scoping review confirms that negative behaviors among nurses are not isolated incidents but persistent and systemic phenomena with serious consequences for workforce sustainability and patient safety. Nurse managers and healthcare leaders must move beyond reactive interventions and invest in long-term strategies that promote psychological safety, conflict literacy, and ethical leadership. Institutional commitment to zero tolerance, structured reporting, and leadership development is no longer optional—it is an urgent imperative. 

Healthcare institutions must recognize negative workplace behaviors not merely as interpersonal issues but as systemic threats to workforce sustainability and care quality—requiring structural responses, not just individual interventions.

## Figures and Tables

**Figure 1 healthcare-13-02079-f001:**
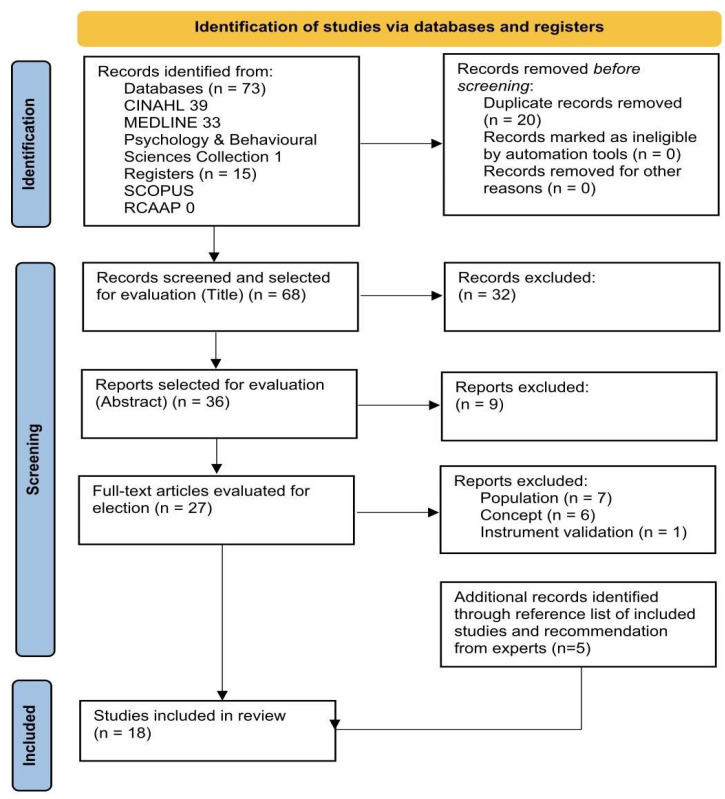
The PRISMA flowchart diagram [29].

**Figure 2 healthcare-13-02079-f002:**
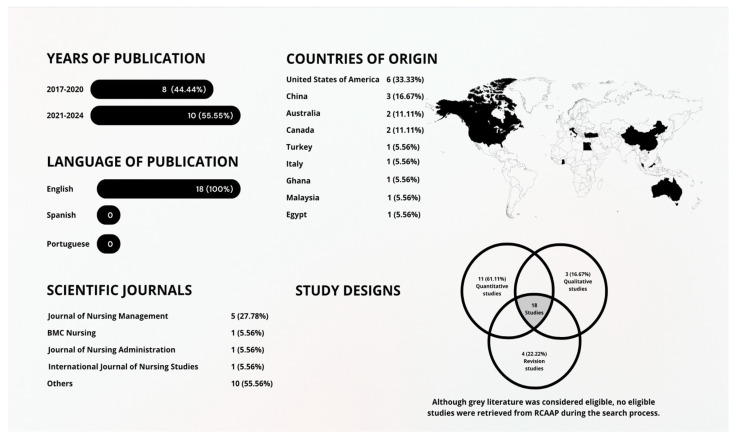
Main characteristics of the selected studies.

**Figure 3 healthcare-13-02079-f003:**
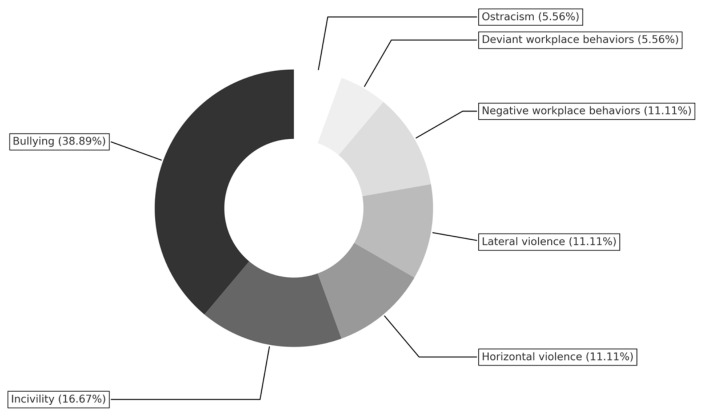
Prevalence of negative behaviors across the included studies.

**Figure 4 healthcare-13-02079-f004:**
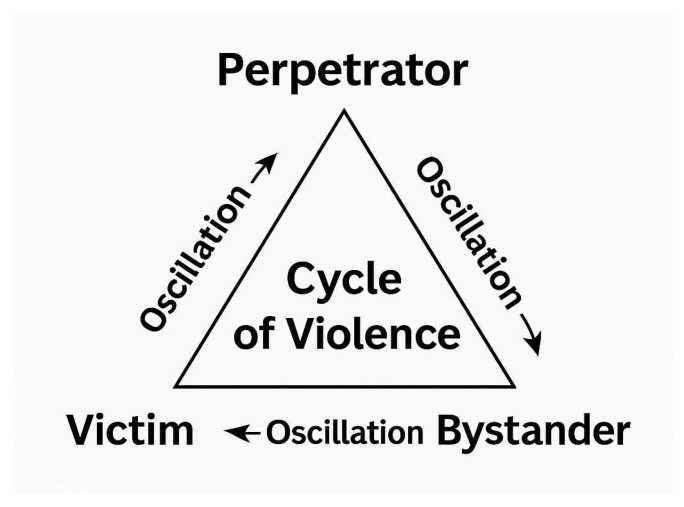
The cycle of violence.

**Table 1 healthcare-13-02079-t001:** Research Question and PCC.

Review Question	PCC	Content
What is the scientific evidence on negative behaviors among nurses in healthcare organizations?	P (population)	Licensed registered nurses working in healthcare organizations.
	C (concept)	Negative workplace behaviors, including but not limited to incivility, bullying, and lateral violence.
	C (context)	Healthcare institutions such as hospitals, clinics, and other formal care settings where nursing practice occurs.

**Table 2 healthcare-13-02079-t002:** Research terms and results.

Databases (Results)	Research Terms	Research Date
CINAHL (39)	(Negative Behavio*)) AND (MM Nurses +))	16 December 2024
MEDLINE (33)	(Negative Behavio*)) AND (MM Nurses +))	16 December 2024
Psychology & Behavioral Sciences Collection (1)	(Negative Behavio*)) AND (DE “Nurses” OR DE “ADVANCED practice registered nurses” OR DE “ASSOCIATE degree nurses” OR DE “CARDIOVASCULAR nurses” OR DE “CLINICAL nurse leaders” OR DE “COMMUNITY health nurses” OR DE “CRITICAL care nurses” OR DE “EMERGENCY nurses” OR DE “EXPATRIATE nurses” OR DE “FAMILY nurses” OR DE “FRONTLINE nurses” OR DE “GAY nurses” OR DE “HOSPICE nurses” OR DE “HOSPITAL nursing staff” OR DE “JEWISH nurses” OR DE “LGBTQ+ nurses” OR DE “MALE nurses” OR DE “MEDICAL-surgical nurses” OR DE “MILITARY nurses” OR DE “NATIVE American nurses” OR DE “NEUROLOGICAL nurses” OR DE “NURSE administrators” OR DE “NURSING consultants” OR DE “ONCOLOGY nurse navigators” OR DE “OPERATING room nurses” OR DE “PALLIATIVE care nurses” OR DE “PEDIATRIC nurses” OR DE “PRACTICAL nurses” OR DE “PRISON nurses” OR DE “PSYCHIATRIC nurses” OR DE “PUBLIC health nurses” OR DE “RURAL nurses” OR DE “SEXUAL assault nurse examiners” OR DE “VISITING nurses”)	16 December 2024
SCOPUS (15)	KEY (“negative behavior*) AND KEY (“nurses”) AND SUBJAREA (NURS)	16 December 2024
RCAAP (0)	(Negative Behavio*) AND (Nurses)	16 December 2024

**Table 3 healthcare-13-02079-t003:** (**a**) Study identification and conceptual focus. (**b**) Methodological features and main findings.

(a)				
Author (Year)	Country	Data Collection Instruments	Type of Negative Behavior	Purpose
Sauer & McCoy (2017) [35]	United States	Negative Acts Questionnaire-Revised, 36-item Short Form Health Survey, Perceived Stress Scale, and 25-item Resilience Scale	Bullying	Examines the impact of resilience on the health effects of bullying among nurses.
Olender (2017) [32]	United States	Caring Factor Survey-Caring of the Manager and the Negative Acts Questionnaire-Revised	Bullying	Examines the relationship between nurse manager caring and perceived exposure to workplace bullying across healthcare settings.
Ayakdaş & Arslantaş (2018) [45]	Turkey	Workplace psychological violence behavior assessment and development scale	Colleague violence	Determines the prevalence of colleague violence among nurses.
Smith et al. (2018) [33]	United States	Workplace Incivility Scale and Practice Environment Scale of Nurse Work Index	Incivility	Determines whether coworker incivility is associated with the nurse work environment and levels of autonomy.
Hawkins et al. (2019) [44]	Australia	Not applicable	Negative workplace behaviors	Synthesizes evidence on negative workplace behaviors experienced by new graduate nurses in acute care and discusses implications for nursing.
Bambi et al. (2019) [38]	Italy	Negative Acts Questionnaire-Revised	Lateral violence and bullying	Investigates the prevalence and risk factors of lateral violence and bullying among Italian nurses across various work settings.
Edmonson & Zelonka (2019) [34]	United States	Not applicable	Bullying	Discusses forms, perpetrators, and contributing factors of nurse bullying; its impact on clinical and financial outcomes; and strategies for prevention.
Anusiewicz et al. (2019) [37]	United States	Not applicable	Bullying	Conducts a concept analysis of workplace bullying to clarify its definition and distinguish it from other forms of workplace violence.
Krut et al. (2021) [40]	Canada	Not applicable	Horizontal violence	Explores the impact of horizontal violence on graduate nurses within their first year of practice.
Xiaolong et al. (2021) [42]	China	Interpersonal Conflict at Work	Incivility	Investigates how coworker and supervisor incivility moderate the relationship between polychronicity and psychological well-being.
Trépanier et al. (2021) [39]	Canada	Negative Acts Questionnaire-Revised	Bullying	Examines how social support and recognition moderate the longitudinal relationship between workload and bullying among nurses.
Peng et al. (2022) [43]	China	Negative Acts Questionnaire-Revised	Horizontal violence	Analyzes the prevalence of horizontal violence among nurses in China and examines the influence of head nurse’s caring and group behavior.
Farrell (2022) [36]	United States	Not applicable	Incivility and bullying	Reviews the impact of incivility in nursing, including effects on nurses, patients and organizations and discusses strategies and national calls to address it.
Mansor et al. (2022) [48]	Malaysia	Not applicable	Deviant workplace behavior	Analyzes deviant workplace behavior and its antecedents among Malaysian nurses in public hospitals.
Hawkins et al. (2022) [1]	Australia	Not applicable	Negative workplace behaviors	Explores nurses’ experiences and perceptions of workplace behaviors in regional acute care settings.
Lu et al. (2022) [41]	China	Workplace Psychologically Violent Behaviors Instrument	Bullying	Explores the association between workplace bullying and suicidal ideation and attempts.
Dapilah & Druye (2024) [46]	Ghana	Negative Acts Questionnaire-Revised and Depression Anxiety Stress Scale version 21	Bullying	Examines the relationship between bullying, depression, and intention to leave the profession.
Elliethey et al. (2024) [47]	Egypt	Work ethics questionnaire, counterproductive work behavior scale, and workplace ostracism scale	Ostracism and counterproductive work behaviors	Examines the relationship between work ethics and counterproductive work behaviors, with ostracism as a mediator.
(**b**)				
**Author (Year)**	**Type of Study**	**Sample Size**	**Key Findings**	**Limitations**
Sauer & McCoy (2017) [35]	Cross-sectional descriptive study	345 nurses	Workplace bullying is associated with poorer physical and mental health, reduced quality of life, and compromised patient care.	Single-state sample with low response rate; potential self-report bias; general health measure may lack sensitivity to specific stressors.
Olender (2017) [32]	Descriptive correlational study	156 nurses	Higher nurse manager caring is associated with lower exposure to workplace bullying; gender, work environment, and workload influence this relationship.	Small final sample size, lack of data on participants’ religiosity, and frequency of interactions with nurse managers may have influenced results.
Ayakdaş & Arslantaş (2018) [45]	Cross-sectional study	779 nurses	Approximately 50% of nurses reported exposure to colleague violence, attributed to factors such as jealousy, educational level, rivalry, inexperience, and workload.	Self-reported data, short duration, and use of basic statistical analysis may have affected the accuracy and depth of findings.
Smith et al. (2018) [33]	Correlational and cross-sectional study	233 nurses	Coworker incivility was inversely associated with the nurse work environment, with nurse manager qualities identified as the key influencing factor.	Causal inference is limited by cross-sectional design; results may not generalize beyond magnet hospitals; low response rate and same-source bias may have influenced findings.
Hawkins et al. (2019) [44]	Integrative review	Not applicable	Lack of conceptual clarity around negative workplace behaviors; leadership style plays a central role in shaping workplace culture.	Exclusion of grey literature, inclusion of studies with small or biased samples and self-reported data may affect generalizability; interpretation requires caution due to methodological variability.
Bambi et al. (2019) [38]	Cross-sectional study	930 nurses	Bullying and lateral violence were present across all studies settings.	Low response rate and potential self-selection bias may have led to overestimation; survey fatigue and technical issues may have affected participation.
Edmonson & Zelonka (2019) [34]	Literature review	Not applicable	Bullying is a systemic issue contributing to nurse turnover, poor work environments, patient risk, and financial losses; addressing it requires cultural change and strong policies.	Use of self-reported data and focus on specific settings may introduce bias and limit generalizability across healthcare contexts.
Anusiewicz et al. (2019) [37]	Concept analysis study	Not applicable	Workplace bullying involves repeated negative behaviors within power hierarchies, leading to adverse outcomes for nurses, patients, and organizations; linked to poor leadership and resource scarcity.	Non-exhaustive search excluding non-English studies; variability in definitions and measures hinders synthesis; focus limited to bullying targeting newly licensed nurses.
Krut et al. (2021) [40]	Descriptive qualitative study	8 nurses	Horizontal violence is prevalent in nurses’ first year of practice, contributing to psychological harm and early career attrition; themes include toxic culture, fear, and isolation.	A small, heterogeneous sample of eight nurses from one Canadian hospital limits generalizability; variation in time since exposure to horizontal violence may have influenced perceptions.
Xiaolong et al. (2021) [42]	Cross-sectional, correlational study	260 nurses	Polychronicity is associated with higher job engagement and performance; however, coworker incivility weakens its positive effect on psychological well-being.	Use of a single self-reported questionnaire may introduce recall and social desirability biases; caution is needed in interpreting results.
Trépanier et al. (2021) [39]	Longitudinal study	279 nurses	Workload predicted bullying over time only when social support and job recognition were low; high recognition was associated with reduced bullying.	Findings are based solely on targets’ perspectives; lack of perpetrator data limits understanding of mechanisms; conducted in one Canadian province, limiting generalizability.
Peng et al. (2022) [43]	Cross-sectional study	1942 nurses	Horizontal violence is prevalent among Chinese nurses; head nurse’s caring and positive group climate act as protective factors.	Self-reported data and 6-month recall period may introduce bias; limited to seven hospitals in Hubei; cross-sectional design prevents causal inference.
Farrell (2022) [36]	Opinion article	Not applicable	Workplace bullying is associated with high workload and low social resources, supporting its interpretation as a strain reaction to adverse work conditions.	Not stated in the study.
Mansor et al. (2022) [48]	Cross-sectional study	387 nurses	Malaysian nurses reported positive affectivity but also high emotional exhaustion, indicating a dual emotional experience in the workplace.	Cross-sectional design limits causal inference; samples restricted to public hospitals in peninsular Malaysia; only two personality traits were examined.
Hawkins et al. (2022) [1]	Mixed-methods sequential explanatory study	13 nurses	Identified the core category “a conflicted tribe under pressure”, with five subcategories: belonging to the tribe, ‘it’s a living hell’, zero tolerance—‘it’s a joke’, conflicted priorities, and shifting the cultural norm.	A convenience sample of 13 RNs from one region in Australia limits generalizability; exclusion of other nursing roles may have restricted the scope of perspectives.
Lu et al. (2022) [41]	Cross-sectional study	1901 nurses	Workplace bullying affected 30.6% of nurses and was independently associated with higher rates of suicidal ideation (16.8%) and attempts (10.8%); risk increased cumulatively with number of bullying subtypes.	Cross-sectional design limits causal inference; self-reported data may introduce bias; suicide assessed with two items; sample limited to tertiary hospitals; COVID-19-related variables were not measured.
Dapilah & Druye (2024) [46]	Cross-sectional study	315 nurses	Workplace bullying was positively associated with nurses’ intention to quit and higher levels of depression.	Not stated in the study.
Elliethey et al. (2024) [47]	Descriptive correlational study	369 nurses	Workplace ostracism negatively influences nurses’ attitudes, leading to adverse behavioral outcomes.	Use of self-reported data may introduce social desirability bias and affect the accuracy of perceived relationships; generalizability is limited.

**Table 4 healthcare-13-02079-t004:** Victims and perpetrators profile.

Victims	Perpetrators
Early-career professionals	Hold positions of authority
Newly hired or in lower hierarchical positions	Exercise power (formal, informal, perceived, or contested)
Lack of support networks	Justify behaviors as “tough love”
Female nurses	May feel threatened by competent peers
Professionals with higher academic qualification	Avoidance-based coping strategies

**Table 5 healthcare-13-02079-t005:** Comparative definitions and characteristics of negative behaviors in nursing.

Term	Definition	Frequency/Intensity Threshold
Incivility	Low-intensity, ambiguous hostile acts; lack of courtesy, disrespect, often subtle but impactful on interpersonal relations.	Sporadic to frequent; lower intensity.
Ostracism	Perceived social exclusion via subtle behaviors (avoiding eye contact, withdrawal, marginalization).	Sporadic but highly impactful.
Colleague Violence	Psychological and other hostile behaviors within teamwork, affecting mental health and care quality.	Frequency varies; includes verbal and psychological abuse.
Horizontal/Lateral Violence	Negative behavior between peers (same hierarchical level); overt (verbal harassment) or covert (social exclusion); repeated and cumulative.	Often daily/frequent; systematic/repeated, cumulative effect emphasized.
Bullying	Repeated, intentional negative behavior with power imbalance, creating hostile environment; verbal aggression, social exclusion.	≥6 months exposure; weekly occurrence.

**Table 6 healthcare-13-02079-t006:** Risk and protective predictors.

Risk Predictors	Protective Predictors
High workload	Favorable nursing practice environments
Shortage of resources	Adequate institutional support
Ineffective communication	Effective communication Working in a hospital environment
Day shifts	Work ethic
Poor leadership	Positive coping strategies
Rural or remote contexts COVID-19 pandemic impact	Professional autonomy

**Table 7 healthcare-13-02079-t007:** Impacts of negative behaviors.

Nurses	Patients	Organization
Emotional strain	Compromised patient safety	Destabilized team dynamics
Somatic symptoms	Increased risk of medication errors	Staffing shortages
Post-traumatic syndrome	Increased number of patient falls	Increased turnover rates
Conflicting personality traits	Increased adverse event risk	Higher absenteeism
Suicidal ideation and attempts		Job dissatisfaction
Spillover into personal life		Economic burden

**Table 8 healthcare-13-02079-t008:** Strategies to prevent and address negative behaviors among nurses.

Strategy Type	Actions
Leadership Approach	Adopt empathetic, supportive, and communicative leadership styles; avoid authoritarian, top-down management styles that exacerbate power imbalances and conflict.
Early Intervention and Prevention	Identify and intervene in negative behaviors early; eliminate behavioral triggers.
Educational and Development	Provide ongoing professional development for leaders and frontline staff; implement cognitive rehearsal (scenario review, role-play); offer continuing education programs on behavioral awareness.
System-Level Interventions	Rotate team composition regularly to reduce tension; increase access to occupational psychologists; establish anonymous reporting systems; implement multi-level organizational interventions.
Policy Implementation	Acknowledge the existence of negative behavior in the nursing workplace; address situational aggravators (e.g., overload, stress, fatigue); train managers in communication and collaboration skills; establish and enforce a zero-tolerance policy for persistent offenders; encourage mutual accountability.

## Data Availability

All data analyzed during this study are included in this published article.

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
