# Peer review of "Unveiling the Dark Side of Negative Behaviors Among Nurses and Their Implications in Workforce Well-Being and Patient Care: A Scoping Review"

_healthcare, 2025, doi:10.3390/healthcare13162079_

Round 1
Reviewer 1 Report (New Reviewer)
Comments and Suggestions for Authors
Thank you for the opportunity to review this thoughtful and timely manuscript. Your work addresses an important issue in nursing and offers valuable insights. I’ve shared some suggestions below that I hope will support and strengthen your already meaningful contribution.
Introduction
The introduction is well-written and provides the necessary background to frame the study. It effectively sets the stage for the reader and clearly communicates the importance of the topic. Great work here!
Methods
There are several areas where additional clarity and refinement could strengthen this section:
- Inclusion Criteria: It’s unclear whether the included studies focused solely on nurses or if mixed samples (e.g., nurses, ancillary staff, providers) were also considered. Clarifying this distinction would help readers better understand the scope of your synthesis.
- Clinical vs. Faculty Nurses: Was it required that participants be clinical nurses? How were studies involving nurse faculty or nurse managers handled? This is an important distinction that should be made explicit.
- Search Strategy: The current search strategy may not be comprehensive enough to capture all relevant studies. For example, I would have expected to see studies using validated instruments such as the Nursing Incivility Scale or the Negative Behaviors in Healthcare Survey included and they appear to be missing. This may be due to the search terms being limited to “nurses” without including broader or truncated terms like “nursing,” “nurs*,” or “registered nurs*.” Expanding the search terms will ensure a more complete capture of the literature. Interestingly you utilize “nursing” as one of your key terms for this manuscript but did not include it as part of your search strategy.
- Hand Searching: It’s also worth noting that hand searching of reference lists from included articles was not conducted. This is a widely recommended strategy to enhance completeness and could be a valuable addition.
Results
This section presents rich and meaningful findings. With a few structural adjustments, the clarity and impact could be significantly enhanced:
- Subheadings for Clarity: Many parts of the results would benefit from revised or additional subheadings. For example:
- In section 3.2.1, consider adding a subheading to clearly mark the shift from the cycle of violence to sample characteristics.
- Subheadings that distinguish risk and protective factors and further differentiate between individual-level and organizational-level factors, would improve readability and help guide the reader through the findings.
- In the section describing the impact of negative behaviors, more subheadings would help organize the content and make the consequences more visible.
- Typographical Corrections:
- Line 427: “38,8%” should be corrected to “38.8%”
- Lines 431–432: Replace the comma with a period within the reported percentages.
- Organizational Context: There is a strong emphasis on the role of the nurse manager, but less attention to the broader organizational systems that shape their capacity to respond. Since nurse managers operate within these systems, reorganizing the results to highlight this dynamic would provide a more balanced view.
- Burnout: Burnout is mentioned in the discussion as a consequence of negative behaviors, but it is not clearly presented in the results. Including it in the results would help readers connect the findings more cohesively.
- Conciseness: The manuscript is quite detailed, which is valuable, but some sections could be streamlined to improve flow and maintain reader engagement.
Tables
- Table 5 Revision: Consider revising Table 5 to include the instrument used to measure negative behaviors (e.g., Nursing Incivility Scale, Negative Behaviors in Healthcare Survey). This would provide important context for interpreting the findings.
- Splitting Table 5: To improve readability, it may be helpful to split Table 5 into two separate tables:
- Table 5a: Author, country, instrument, purpose, and type of negative behavior
- Table 5b: Sample demographics/sample size, key findings, and limitations
This restructuring would make the information more digestible and easier to navigate for readers.
Author Response
Found in the document

Reviewer 2 Report (New Reviewer)
Comments and Suggestions for Authors
Title:
Unveiling the Dark Side of Negative Behaviors among Nurses and Their Implications in Workforce Well-Being and Patient Care: A Scoping Review
Abstract
-
The number of thematic domains mentioned in the abstract does not align with the listed items; six domains are stated but only five are provided. This discrepancy should be corrected for clarity and consistency.
Introduction
-
The introduction incorporates a large volume of theories, concepts, and classifications without clear differentiation or logical categorization, which may overwhelm and confuse readers.
-
Multiple definitions of negative behaviors (e.g., ILO, Nemeth, general terms) are presented but lack comparative analysis or synthesis, leading to conceptual ambiguity.
-
Although the absence of a coherent theoretical framework is noted at the end of the introduction, this gap could be more explicitly and clearly articulated to highlight the specific shortcomings of previous research.
-
The concept of “silent epidemics” is introduced without sufficient explanation; this specialized term is only briefly mentioned. It would benefit from further elaboration, ideally with examples or contextualization relevant to nursing.
-
The inconsistency among definitions of negative behaviors contributes to conceptual confusion and would benefit from an integrated or comparative discussion.
Methods
-
The rationale for selecting a scoping review design should be explicitly justified. The characteristics and methodological features of scoping reviews ought to be described in the methods section. Based on the presentation of results, this study reads more like a systematic review or narrative review, which should be clarified.
-
The first paragraph of the methods is excessively long and could be condensed for conciseness.
-
There is an inconsistency regarding grey literature: the RCAAP database is mentioned as a source for grey literature, but later it is stated that grey literature was excluded due to lack of eligible studies. To address this contradiction, consider adding a sentence in the Eligibility Criteria section such as:
“Although grey literature was considered eligible, no eligible studies were retrieved from RCAAP during the search process.” -
While the included languages (English, Portuguese, Spanish) are stated, there is no mention of exclusion criteria related to articles published in other languages; this should be clarified.
-
The method for screening studies and assessing their quality or eligibility should be described to ensure methodological transparency.
Results
-
As a scoping review, results typically summarize findings and present them visually through charts or diagrams. The current presentation is overly detailed and text-heavy, which may be tedious for readers.
-
The tone of the results is heavily negative and somewhat discouraging. As a nursing expert, I found this disheartening. It is recommended to present findings more succinctly and balance the narrative where possible.
-
It would improve the manuscript to include some quantitative summary, such as the proportion of studies reporting on different types of negative behaviors (e.g., what percentage addressed violence, bullying, incivility, etc.) within the 18 included studies.
Discussion
-
The discussion is thorough and provides a comprehensive overview of negative behaviors in nursing. However, it is excessively lengthy and would benefit from condensation to enhance readability.
-
Limitations and conclusions sections should also be more concise to maintain focus and impact.
The manuscript needs extensive English language editing
Author Response
Found in the document.

Round 2
Reviewer 1 Report (New Reviewer)
Comments and Suggestions for Authors
Thank you for the opportunity to review your revised manuscript. It is clear prior feedback was thoughtfully addressed and incorporated. One additional consideration: the limitations section does not mention that the included studies were not critically appraised for methodological quality. This is important to clarify, as readers may otherwise assume that the reported results were drawn from high-quality, rigorous studies-which may not be the case. Including this information helps to ensure the transparency and strengthens the overall integrity of the manuscript.
Author Response
Please see the attachment.

Reviewer 2 Report (New Reviewer)
Comments and Suggestions for Authors
My comments have been addressed satisfactorily.
Author Response
For Reviewer 2, no changes were required as the reviewer confirmed that all previous comments had been addressed satisfactorily.
This manuscript is a resubmission of an earlier submission. The following is a list of the peer review reports and author responses from that submission.
Round 1
Reviewer 1 Report
Comments and Suggestions for Authors
Thank you for the opportunity to evaluate this manuscript titled: Negative Behaviors in Nursing: A Scoping Review. I think this is a promising piece and will be a valuable contribution to nursing literature but needs major revisions, particularly on deeper discussion of practical strategies and research gaps, clarification of theoretical framing and so on. More specifically:
1/ The title need to revise and better capture the essence of the manuscript with implications for workforce well-being and patient care.
2/ Add brief justification for scoping review vs. systematic review and meta analysis. Further justification on the selected databases used.
3/ In Introduction, add explanation of theoretical framing of this topic within the broader aspect of workforce well-being and patient care.
4/ The transition from "conflict as harmful" to "conflict as productive" (Lines 41–44) could be smoother. Perhaps clarify how this relates to negative behaviors.
5/ Define "silent epidemics" (Line 47) early on to avoid ambiguity.
6/ In the methods section, specify why the time frame (2017–2024) was chosen beyond "terminological refinement" (Lines 171–174). Was this due to seminal works (e.g., Nemeth et al., 2017)?
7/ Please clarify how grey literature was handled (Line 186). Only RCAAP was mentioned, but no grey literature was included in the final selection.
8/ In the results section, there are repetitive statements, perhaps merge them e.g., "diversity of instruments and domains" appears twice in the Abstract and Introduction.
9/ There are terminological Issues that needs to be addressed. The manuscript uses multiple terms (e.g., "lateral violence," "horizontal violence," "peer violence") interchangeably. A table comparing definitions/frequency thresholds (e.g., bullying ≥6 months vs. incivility as low-intensity) would aid clarity.
In addition, please address contradictions throughout the manuscript. For example, resilience is initially dismissed as ineffective (Lines 535–537) but later noted as part of coping strategies (Lines 460–470).
10/ Please strengthen on the theoretical framework used to guide this review. The reliance on Nemeth et al. (2017) is justified but could be strengthened by linking to broader theories (e.g., Power-Conflict Theory, Social-Ecological Model).
11/ Provide a brief justification on the exclusion of non-English/Portuguese/Spanish studies (Lines 731–732). Could this introduce language bias?
12/ In the discussion section, could you discuss how conflicting findings (e.g., prevalence rates ranging from 27% to 64.4%) were reconciled.
13/ In the implications and recommendations section, please expand on "zero-tolerance policies" (Line 705). What specific interventions (e.g., cognitive rehearsal, anonymous reporting) are most evidence-based? Furthermore, try to address barriers to implementation (e.g., staff shortages, Line 714).
14/ The call for longitudinal studies (Lines 749–750) is apt. Add specific metrics (e.g., turnover rates, patient outcomes) for future studies to track. Suggest mixed-methods approaches to explore cultural/organizational nuances.
Some minor comments for your further consideration:
1/ The formatting could be better: Standardize instrument names (e.g., "Negative Acts Questionnaire-Revised" vs. "NAQ-R").
2/ Check reference consistency (e.g., Hawkins et al. cited as 2023 in text but 2021 in references).
3/ Grammar check, for example, Line 96: "repercussions that go beyond the individual and have a direct impact" → "repercussions extend beyond individuals, directly impacting..."
As well as Line 535: "Resilience, in this context, was not an effective protective factor" → "Resilience did not significantly mitigate effects."
4/ Proofreading for redundancy and consistency.
With these revisions, I look forward to evaluating a much improved version.
Comments on the Quality of English LanguageAs listed in comments to authors.
Reviewer 2 Report
Comments and Suggestions for Authors
Dear Authors,
the comments in the annex file.
Best

Round 2
Reviewer 1 Report
Comments and Suggestions for Authors
Thank you for thoroughly addressing my previous comments and taking up the suggestions for improvement. The manuscript reads much better and is ready for publication. All the best to the authors.
Author Response
Thank you so much for your comments
Reviewer 2 Report
Comments and Suggestions for Authors
The comments in the annex file

Native review required